# Models and Interventions to Promote and Support Engagement of First Nations Women with Maternal and Child Health Services: An Integrative Literature Review

**DOI:** 10.3390/children9050636

**Published:** 2022-04-28

**Authors:** Catherine Austin, Danny Hills, Mary Cruickshank

**Affiliations:** 1School of Health, Federation University, Mount Helen, VIC 3350, Australia; d.hills@federation.edu.au (D.H.); m.cruickshank@federation.edu.au (M.C.); 2Health Innovation and Transformation Centre, Federation University, Mount Helen, VIC 3350, Australia

**Keywords:** First Nations, maternal and infant health, social determinants of health

## Abstract

Background: Studies show that participation in maternal and child health (MCH) services improves health outcomes for First Nations families. However, accessing MCH services can be associated with fear, anxiety, and low attendance at subsequent appointments. Objective: To identify the existing knowledge of models/interventions that support engagement of First Nations women with MCH services in the child’s first five years. Methods: An integrative review was undertaken of full-text, peer-reviewed journal articles and grey literature, which were analysed to identify barriers and enabling factors that influenced the engagement of First Nations families with MCH services. Results: Enabling factors that influenced the engagement with MCH services included service models/interventions that are timely and appropriate, and effective integrated community-based services that are flexible, holistic, culturally strong, and encourage earlier identification of risk and further assessment, intervention, referral, and support from the antenatal period to the child’s fifth birthday. Barriers to engagement included inefficient communication, lack of understanding, cultural differences between the client and the provider, poor continuity of care, limited flexibility of service delivery to meet individual needs, and a health care model that does not recognise the importance of the social determinants of health and wellbeing. Discussion: Timely, effective, holistic engagement with First Nations women during their child’s first 2000 days, which respects their culture and facilitates genuine partnerships built on co-design and shared decision making with the indigenous community, needs to be an essential part of the MCH service model if health care providers seek to practice within First Nations communities. Conclusion: Improving engagement with MCH services is important for First Nations families, nursing practice, and public health.

## 1. Introduction

The early years of life are the foundation for lifelong physical, social, spiritual, and emotional wellbeing [1,2]. The first 2000 days, the antenatal period to the child’s fifth birthday, comprise the critical period of foetal and child development, which forms the foundation for all subsequent development and health throughout the child’s lifetime [3]. The period of early childhood also represents a critical window of opportunity, where optimal health and educational interventions can provide benefits that extend across the life course [3,4,5,6].

The United Nations General Assembly [7] acknowledges the rights of children ‘to the highest attainable standard of health’ with a focus on preventative and primary health care for children, prenatal and postnatal health care for mothers, and diminishing infant and child mortality. The need for improved child health care for First Nations populations is evident by ongoing disparities in child health among the indigenous and tribal populations in Australia, Canada, New Zealand, and the United States [8]. Ensuring First Nations women have access to appropriate health care in the antenatal and postnatal period is vital for the long-term health outcomes for the mother and her child [9]. First Nations peoples, also referred to as indigenous peoples, are the direct descendants from the original known inhabitants of a geographic region [10].

Many First Nations families deal with ongoing stressors from the intergenerational trauma of the impacts of colonisation [8]. This can manifest in psychological distress or grief, smoking, alcohol and drug misuse, mental illness, or violence and consequently may affect their ability to nurture children [10]. However, First Nations families have strong bonds with their immediate family members and extended families, which influences the cultural norms governing their child rearing practices [10]. These strong family bonds provide opportunities for Maternal and Child Health (MCH) services to support First Nations families and address upstream social determinants of First Nations children’s health and wellbeing [10,11].

In Australia, MCH nurses, also referred to as ‘child and family nurses’, are uniquely placed in the community to influence the shape of this critical period in a child’s life [6]. Participation in the MCH service provides the opportunity to identify, or prevent, health risks to children and their families [1,2]. Women are referred to MCH services by midwives from maternity services after birth and are offered a schedule of routine monitoring of child development, parenting support, health promotion services, and early identification of family needs and responses to these needs until the child is at least five years old [6]. Despite the aims of universal service provision, populations such as some indigenous and culturally and linguistically diverse (CALD) families do not engage with services [12] or do not sustain visits [13], and economically disadvantaged families are less inclined to access services [14]. For example, although 95–98 per cent of Victorian women with new-born children are being visited at home within two weeks of birth [6], there are critics of the current Victorian universal MCH service model, especially in relation to the engagement of Aboriginal and Torres Strait Islander families [6,12,14]. This is largely due to the consistently lower participation rates of indigenous children compared to non-indigenous children at all 10 Key Age Stage (KAS) consultations provided within the universal MCH service since the inception of the KAS model in 2009 [6]. This gap increases from the initial home visit to the eight-month consultation, indicating that First Nations women in Victoria are disengaging with the MCH service shortly after their initial enrolment in the service [6]. This brings into effect an ‘inverse care law’, suggesting that those who need more intense, high-quality care are least likely to receive it [14]. It is also well recognised that there is poor health among indigenous families in countries with a history of colonisation, such as Australia, New Zealand, and Canada, and access to services is adversely affected by historical and contemporary social determinants of health, such as the distribution of power, influence, wealth, and income [8]. Indigenous parents in particular require support services that are culturally strong, timely and appropriate, and holistic to strengthen their families’ health outcomes [1,2,15]. Ensuring the optimal design of a model that promotes and supports the engagement of First Nations families and their access to MCH services in the period from the child’s birth to five years of age is impeded by the dearth of information on the relationship between specific services and children’s health outcomes [16]. The aim of this review, therefore, is to identify the factors that support or hinder the engagement of First Nations women with children from birth to five years of age and their access to MCH services, and the improvements that could be made to enhance engagement and access for First Nations women in relation to these services.

## 2. Materials and Methods

### 2.1. Search Criteria

An integrative literature review entails undertaking a systematic search, critique, and summary of relevant literature [17,18]. The inclusion criteria for this review were full-text, peer-reviewed journal articles and grey literature of relevant studies that investigated models and interventions that aim to promote and support the engagement of First Nations families and their access to MCH services in the period from the child’s birth to five years of age. The exclusion criteria were study protocols or policy perspectives, which discussed risk factors for specific diseases or practices, descriptions or evaluations of interventions or programs that exclusively focused on the antenatal period, or reports of pregnancy or birth outcomes.

### 2.2. Search Strategy and Outcomes

Searches were conducted in Medline, PsychInfo, CINAHL, and Scopus databases, and relevant government publications from January 2011 to April 2021 were reviewed for inclusion. Search terms, singular and in combination, of the search terms ‘matern*’, ‘child*’, ‘famil*’, ‘postnatal’, ‘service*’, ‘care’, ‘health’, ‘model*’, ‘intervention*’, ‘approach*’, ‘indigen*’, ‘aborig*’, ‘torres strait’, ‘first nation*’, ‘native’, ‘engag*’, ‘interact*’, ‘uptake*’, ‘continu*’, ‘access*’, were included, in conjunction with the terms ‘and’ and ‘or’.

The initial database search identified 732 records, with an additional 99 records identified from grey literature on Google Scholar (47), hand searching and snowballing from the reference lists of included articles (43), and utilising peer referral (9). These searches yielded a total of 831 records. After the preliminary screening that entailed the removal of duplicates (30), 801 records remained. Following a further screening of the titles and abstracts of articles against the inclusion and exclusion criteria, 652 records were deleted, and 149 records remained. After a secondary screening of full-text articles against the inclusion and exclusion criteria, a further 143 records were excluded, and six records remained. These six records were individually assessed by the authors who helped determine if they met the inclusion criteria, all six being included in the final review. Each stage of the literature search is summarized in Figure 1.

Articles were analysed by authors CA, DH, and MC separately and then collectively, using Braun and Clarke’s six-step process for identifying, analysing, and reporting qualitative research using thematic analysis [19]. Thematic analysis provided an easily interpretable and concise description of the emergent themes and patterns to identify broad concepts of the barriers and enabling factors that influenced access to and engagement of First Nations families with MCH services. The six-step process included familiarising ourselves with the research articles; generating initial codes; searching for themes; reviewing the themes; defining and naming the themes; and producing a report with the themes found within the research articles.

A summary of the analysed studies is presented in Table 1.

## 3. Results

The review of the literature identified a limited number of studies of models that promote and support the engagement of First Nations families and their access to MCH services in the period from the child’s birth to five years of age (Table 1).

All six studies included in the analysis were published between 2012 and 2018. Two were mixed-methods studies with an intervention that aimed to promote and support access to and engagement of First Nations families with MCH services [20,24]; one was a mixed-methods study with no intervention [21]; one was a quantitative study with an intervention [22]; and two were qualitative studies with an intervention [11,23] that aimed to promote and support the engagement of First Nations families with and their access to MCH services. All included studies were either conducted across metropolitan [22], remote [11,23,24], or a combination of metropolitan and remote settings [20,21] in Australian Aboriginal communities.

### 3.1. Enabling Factors That Influenced Access to and Engagement of First Nations Families with MCH Services

In this review of the literature, the gaps in the research evaluating service models and interventions that enabled the engagement of First Nations families with and their access to MCH services were identified. The enabling factors identified from the six included studies were service models or interventions that are timely and appropriate; effective integrated community-based services that were flexible in their approach; holistic service models or interventions; culturally strong service models or interventions; and service models or interventions that encourage earlier identification of the risk and the need for further assessment, intervention, referral, and support from the antenatal period to the child’s fifth birthday (the first 2000 days).

#### 3.1.1. Timely and Appropriate Service Models or Interventions

Our analyses of the studies included in this review revealed that, as MCH services respond to the needs of children and families at risk of vulnerability, many childhood conditions that pose a risk for poor outcomes can benefit from early detection and intervention actions [11,20,21,23,24]. For this reason, there is a need for models or interventions that enable timely and appropriate services for First Nations families from conception to the child’s fifth birthday, such as those identified in all six of the studies included in the review. In particular, the Apunipima Baby Basket program and the Australian Nurse-Family Partnership Program were more successful in supporting access to and engagement of First Nations families with MCH services, as these models or interventions were seen as mutually valuable to both MCH nurses and to First Nations women [11,20,21,23,24].

#### 3.1.2. Effective Integrated Community-Based Services That Are Flexible in Their Approach

The MCH service in Australia is part of a larger service system, which builds on the identification of individual, family, and community needs at the local level. The MCH service has the flexibility to devise and implement innovative service models, which support integration and collaboration of services while maintaining the universal nature of the service. Examples of models that promote service integration include co-location of services, where services are located together; interdisciplinary teams, where knowledge comes from a number of different medical specialties; protocol sharing, where structure and language for files between clients and servers are enabled; and joint service delivery, where two or more organisations come together to deliver a joint service using common assessment frameworks and referral tools. The included studies showed that effective, integrated, community-based services that are flexible in their approach and collaborate with other early years services enabled more effective access to and engagement of First Nations women with children from birth to five years of age with MCH services [11,20,21,22,23,24].

#### 3.1.3. Holistic Service Models or Interventions

The studies included in the review reveal that a strengths-focused approach to raising children enables a shift away from a ‘problem’-focused model to a strength-focused model for families and communities [20,24]. First Nations families frequently experience the strength of a strong bond with their extended families. Service models or interventions should therefore ensure that holistic care is planned around the whole family and supports family lifestyle factors and the interpersonal social determinants of First Nation children’s health and wellbeing [11,20]. Family-centred healthcare, delivered through primary healthcare services for indigenous children in the period from conception to the child’s first five years, is an example of a holistic service model or intervention that provides support for the person as a whole, not just their individual medical needs. The primary care provider considers women’s physical, emotional, social, and spiritual wellbeing using listening, asking, and checking as key skills to be able to provide a holistic service. In the reviewed studies, this holistic approach acted as an enabling factor to support access to and engagement with MCH services [11,20,21,22,23,24].

#### 3.1.4. Culturally Strong Service Models or Interventions

A significant enabling factor identified in the literature, which promotes and supports access to and engagement of First Nations families with MCH services in the period from the child’s birth to five years of age, are models or interventions that facilitate strong relationships between the client and the provider and encourage mutual trust and engagement of First Nations families with MCH services [11,20,21,22,23,24]. This is strengthened when government and policy makers genuinely acknowledge the historical, cultural, and social complexity of First Nation families’ birthing and child-rearing principles and practices, and recognise the importance of culturally strong service models or interventions [11,20,21,22,23,24].

#### 3.1.5. Service Models or Interventions That Encourage Earlier Identification of Risk and Need for Further Assessment, Intervention, Referral, and Support from the Antenatal Period to the Child’s Fifth Birthday (the First 2000 Days)

While professionals working with children in the antenatal period to the child’s fifth birthday (the first 2000 days) have recognised that this is a vital period of child development, the literature reviewed has only recently started to understand the mysteries surrounding the processes by which genes, experiences, and environments interact to influence child development [11,20,21,22,23,24]. This requires a model with a focus that adheres to indigenous methodologies and knowledge, which is holistic and culturally strong, such as the service models and interventions in the included studies, in particular, the 1 + 1 = A Healthy Start to Life Project and the Malabar Community Midwifery Link Service. New knowledge that has emerged from these studies has served to increase experts’ views of the significance of the first 2000 days and of the urgent need to reform the relevant policies, practices, and systems in response to the evidence [11,20,21,22,23,24].

### 3.2. Barriers That Influenced Access to and Engagement of First Nations Families with MCH Services

This review identified barriers that influenced access to and engagement of First Nations families with MCH services, including factors that affected quality of care, service delivery, and outcomes for these families. Addressing these barriers may assist in enabling access to and engagement of First Nations families with MCH services. The barriers identified from the six included studies were: inefficient communication resulting in lack of understanding between the client and the provider; cultural differences between the client and the provider; poor continuity of care between services; lack of flexibility in approach/access to services; and a model that does not recognise the importance of the social determinants of health and wellbeing.

#### 3.2.1. Inefficient Communication Resulting in Lack of Understanding between Client and Provider

A discordance exists between indigenous and non-indigenous views about the role of children and their agency within the family, influencing access to and engagement of First Nations families with MCH services [20,22]. As discussed previously, there are consistently lower participation rates of indigenous children compared to non-indigenous children at all 10 Key Age Stage (KAS) consultations provided within the universal MCH service annually since the inception of the KAS model of care in 2009 [6]. This gap increases between the initial home visit to the eight-month consultation, indicating that First Nations women in Victoria are disengaging with the MCH service shortly after their initial enrolment in the service [6]. Homer et al. discuss the differences in views between indigenous and non-indigenous populations in regard to child rearing that are well documented in countries with a history of European colonisation [22]. The absence of leadership from some First Nations communities to inform a greater respect and understanding of First Nations values and beliefs pertaining to parenting and child rearing, highlighted by Bar-Zeev et al., appears to be a contributing factor [21].

Reconciling divergent views between indigenous and non-indigenous peoples about the role of children in the family, which would require greater understanding of and respect for First Nations values and beliefs pertaining to parenting and child rearing, could be a way of contributing to improved health outcomes, as evidenced in the Australian Nurse-Family Partnership Program [24]. Currently, there is little empirical evidence in the literature pertaining to the implementation of these parenting paradigms.

#### 3.2.2. Cultural Differences between Client and Provider

Culturally inappropriate practices and racist attitudes by clinicians were identified in the included studies as barriers that influenced access to and engagement of First Nations families with MCH services [23]. The small numbers of First Nations staff working in MCH services and perceived racist behaviours by some staff were also barriers to access and engagement [21,23]. Racism was identified as a key determinant of health for First Nations people [23], and the experience of racism by First Nations people can contribute to poor health outcomes [23]. The development of a culturally competent workforce and tools to measure appropriate care in MCH services is required to address this issue [20]. A review of the international Nurse-Family Partnership Program found that program performance is critically dependent on the compatibility between the client characteristics and the program model [24]. It is therefore vital to understand that the nature of the client population and the adversities that they face will affect the likelihood of them accessing and engaging in the services offered to them [24]. Creating a culturally safe practice allows care to be extended to women’s and families’ social and emotional wellbeing [24].

#### 3.2.3. Poor Continuity of Care between Services

Continuity of care, being a philosophy of care, focuses on the quality and consistency of care over time. For providers of integrated systems of care, the ideal is the delivery of a smooth, continuous, and uniformly ‘seamless service’ through integration, coordination, and the sharing of information between different providers. In maternity and MCH services, this refers to service models that incorporate continuity of services and/or continuity of care across antenatal, labour, birthing, and postnatal care. The included studies reported that First Nations women frequently perceive available MCH services as culturally unsafe [11,20,21,22,23,24]. First Nations women reportedly view continuity of care as more culturally safe (than existing siloed care), which can result in a greater uptake in health care across antenatal, labour, birthing, and postnatal care periods [20]. For example, in the study by Homer et al. [22], the focus group findings showed that women felt the service provided ease of access, continuity of care, and trust and trusting relationships.

#### 3.2.4. Lack of Flexibility in Approach/Access to Services

In the included studies, access to pre-conception, antenatal, and postnatal care was found to often be compromised for First Nations women [11,20,21,22,23,24]. Access to effective integrated community-based services that are flexible or variable in their approach to meet the needs of a specific client are often limited for low-socio-economic-status families. In addition, social isolation for many women is exacerbated by new motherhood [22]. McCalman et al. [11] identified home visiting as a key strategy for creating a culturally safe practice while providing flexibility in approach and access to MCH services.

#### 3.2.5. A Model That Does Not Recognise the Importance of the Social Determinants of Health and Wellbeing

The studies included in the review revealed that comprehensive, holistic models of care assisted in enabling access to and engagement of First Nations families with MCH services [11,20,21,22,23,24]. However, the models or interventions that did not incorporate services that address the social determinants of health, namely, economic stability, access to quality education and health care, neighbourhood and environment, and social and community context, were considered to be a significant barrier for engaging First Nations families in MCH services [11,20,21,23,24]. This supports the theory that the gap in life expectancy for First Nations peoples, in comparison to non-First Nations peoples, can be partially attributed to differences in the social determinants of health, including the social and environmental conditions in which people live and work [23]. Examples of these include extreme poverty, welfare dependency, low engagement with work and school, insecure housing, racism, multiple traumas, and domestic violence [23].

## 4. Discussion

The aim of this review was to explore the current literature and identify existing knowledge that can improve First Nations families’ and their children’s access to and participation in MCH services in the period from the child’s birth to five years of age. Although recent adaptations to maternity service models of care show some positive outcomes [23], the included studies indicate that improvements to infant care are required [11,20,21,22,23,24]. The current MCH service systems are often ‘fragmented’ [21]. Consequently, accessing MCH services often results in high levels of fear and anxiety, and low attendance at subsequent appointments among First Nations women [25]. The studies included in the review show that a biomedical model of care underpins most mainstream MCH services, which can be incongruous with traditional indigenous ways of parenting and child rearing [25]. The consequence of the poor understanding of parenting and child-rearing practices of First Nations peoples is that health providers from non-indigenous backgrounds continue to provide health advice and information from their own cultural perspectives [21].

Additionally, the literature shows that the communication on the transfer of mother and infant care from maternity services to MCH services can also be fragmented or inadequate, inconsistent, and ad hoc, resulting in potentially serious clinical consequences for new mothers and their children in this vulnerable period and impacting First Nations families’ and their children’s access to and participation in MCH services [21]. In order to encourage access to and engagement of First Nations families with MCH services, there need to be improvements to the organisation of child health care, including giving priority to the continuity of supportive relationships with parents across the services [20]. As families’ healthcare needs can rarely be met in this modern era by a single professional, multi-dimensional models of continuity have been developed to accommodate the possibility of achieving both ideals simultaneously [20]. Given that health disparities continue to exist for First Nations women and infants, it is imperative to explore the factors that facilitate continuity of care from the antenatal period to the child’s fifth birthday (the first 2000 days of life). Earlier engagement with these families, ideally in the antenatal period before the woman is discharged from maternity services, would facilitate the transition between these services and may build greater trust in the MCH service.

Healthcare professionals are often inadequately trained and underprepared to work cross-culturally, which further compounds the situation [25]. As a result, McCalman et al. [11] report that many First Nations women do not disclose vital health information to healthcare workers with whom they have no relationship. Culturally inappropriate practices were identified in the included studies as a barrier that influenced the engagement of First Nations families and their access to MCH services [23]. Small numbers of First Nations staff working in MCH services and the perceived racist behaviours by some staff contributed to this theme [21,23]. An area requiring reform to encourage engagement with MCH services for this population and assist in closing the gap in health outcomes for First Nations women and their children is the development of a culturally competent workforce, which supports client diversity and complexity [24,26,27]. 

A key gap in the evidence is that there has not been a synthesis of qualitative studies of models of care to help guide MCH practice and innovation for all families, especially those at risk of vulnerability, that is, familial living situations that are considered problematic and in need of professional support [11,20,21,22,23,24]. Future studies may benefit from exploring the significance of the enabling factors identified in the six included studies in this review. For example, Homer et al. [22] reported that the importance of continuity of health care provider was highlighted by study participants, with women describing it as ‘the best part of Malabar’. They valued having a person they could call and receiving care from the same health care provider. Additionally, as highlighted by Barclay et al. [20], models of care or interventions that are more inclined to be successful are those based on earlier engagement of families with MCH services and services with a focus on a continuum of care to alleviate the risk of these families ‘falling through the cracks’.

This review of the literature showed that services must be collaborative, be more ‘connected’, and made easier for families to access to ensure that there is adequate support provided [28]. The continuum of care framework, a concept involving an integrated system of care that follows patients over time through a comprehensive array of health services spanning all levels of intensity of care, may facilitate this [29,30]. However, traditionally, the continuum of care framework has focused on chronological modes of care, without specifically measuring the experienced continuity or the facets of care that translated into intelligible and meaningful care [29,30]. In order to use the continuum of care framework to allow multiple agencies to work together to provide a coordinated, comprehensive service to engage First Nations families, it is imperative that a common understanding of the concept of continuity, as a basis for valid and reliable measurement of practice in different settings, is established [20].

Additionally, Barclay et al. [20] reported that poor knowledge, recognition, and support of diverse culture and child-raising needs in the health system have to be remedied in order for staff to effectively promote health and resilience among parents of children at risk of vulnerability. As urged in the policy reports and charters, worldwide, a concerted effort is required to enhance continuity through a new model of care [20,31,32]. Service design reform to align with the continuity of care model, enabling staff to work alongside First Nations women, their families, and community leaders, could be an important step forward in addressing the disparities in health outcomes between indigenous and non-indigenous children [33,34].

In summary, the literature reviewed showed that timely, effective, holistic engagement with First Nations women in their child’s first 2000 days, which respects their culture and facilitates genuine partnerships built on co-design and shared decision making with the indigenous community, needs to be an essential part of the MCH service model if the health care providers seek to practice within First Nations communities. A critical review of Western models of care that do not support evidence-based best practices for indigenous populations, in conjunction with adopting a strengths-based approach, which respects First Nations peoples’ child-rearing practices and culture, is required to support access to and engagement of First Nations women and their children from birth to five years of age with MCH services [11,20,21,22,23,24].

## 5. Limitations

Although a rigorous and thorough search strategy was used to identify existing knowledge of models and interventions that promote and support the engagement of First Nations families and their access to MCH services in the period from the child’s birth to five years of age, it is possible that this integrative literature review did not identify all relevant studies. The studies were screened individually and assessed by the authors to determine if they met the inclusion criteria. It is possible that relevant models or interventions descriptions or evaluations may have been misclassified. Due to the lack of data internationally, the outcomes of the retrieved studies may not be generalizable to the entire First Nations families worldwide. Additionally, it is impossible to determine any cause-and-effect relationships between the interventions described in the included studies and an improved engagement of First Nations families and their access to MCH services, as the methodological quality of the intervention studies varied considerably.

## 6. Conclusions

The focus of this literature review was to explore and describe the models or interventions that support access to and engagement with MCH services for First Nations women with from birth to five years of age. Persistent disparities in perinatal outcomes between indigenous and non-indigenous families underscore the need to prioritise culturally responsive practices in MCH services. MCH service models and interventions for First Nations families need to facilitate early identification of risk for children and families who may require further assessment, intervention, referral, and support through a transdisciplinary approach, ideally from the antenatal period to the child’s fifth birthday (the first 2000 days). Services that increase accessibility and are designed to support First Nations women during their pregnancy and the postnatal period are likely to have a positive impact. Furthermore, programs are more likely to be accessed by First Nations women if they are designed in a culturally safe and secure space, using a bi-cultural approach that combines the Western biomedical model with indigenous cultural ways of being, doing, and knowing.

A crucial issue in translating the results and recommendations of this review into policy or practice would be to ensure that MCH service models and interventions focus on the issues most relevant to people’s lives, namely, the social determinants of health and wellbeing, and a shared understanding and common language regarding the needs and risks for children and their families. In Australia, given the practice of home visitations by MCH nurses after childbirth, MCH services are well placed to address the inequalities for the most disadvantaged families.

## Figures and Tables

**Figure 1 children-09-00636-f001:**
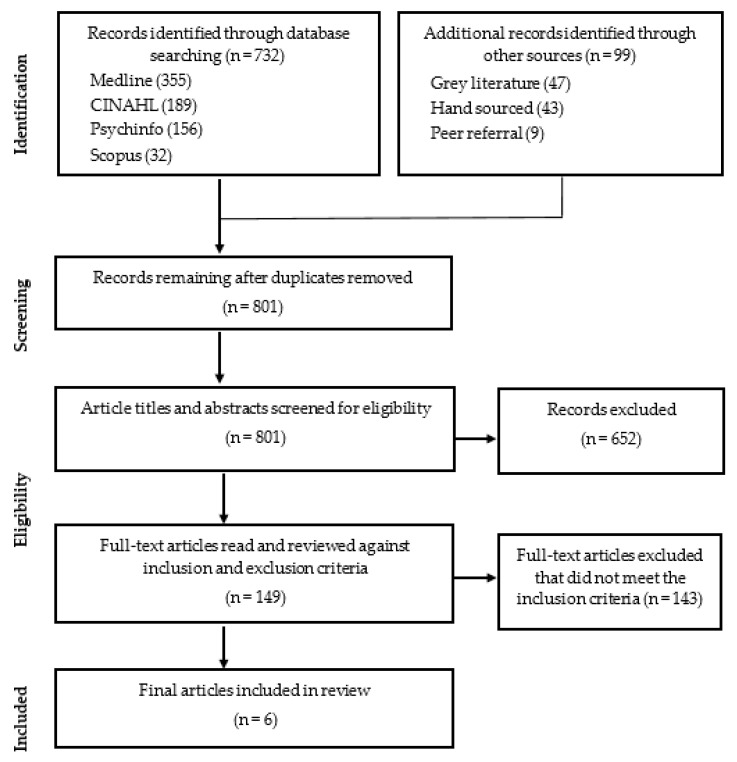
Literature review flow diagram.

**Table 1 children-09-00636-t001:** Models/interventions that promote and support engagement of First Nations families and their access to maternal and child health services from the child’s birth to five years of age.

Author/Date/Title	Sample	Type of Study/ Methodology	Thesis/ Intention of Work	Intervention	Results
Barclay, L.; Kruske, S.; Bar-Zeev, S.; Steenkamp, M.; Josif, C.; Wulili Narjic, C.; Wardaguga, M.; Belton, S.; Gao, Y.; Dunbar, T.; Kildea, S., 2014, [20]. Improving Aboriginal maternal and infant health services in the ‘Top End’ of Australia: synthesis of the findings of a health services research program aimed at engaging stakeholders, developing research capacity and embedding change.	Baseline data study: Data from 412 mothers and their 413 babies who were recruited from two remote study sites over two years (2004–2006) were audited; 120 h of observation of maternal and child health services and 60 semi-structured interviews were conducted in 3 settings with key stakeholders. Epidemiological studies: An epidemiological investigation of 7560 mothers with singleton pregnancies utilizing the Northern Territory perinatal data set that included births occurring between 2003 and 2005 was conducted. Study of out-of-hospital births: Audit of 32 records of women who birthed locally, detailed field notes, stories collected, and unstructured interviews with 7 locally birthing women and 5 of their family members. Parenting study: Longitudinal interviews and observations with 15 women from each field site from pregnancy until their babies were 12 months of age. Discussions were held with women and family members and narratives collected. Impact of colonisation of health care in the Northern Territory: An Aboriginal PhD candidate with Aboriginal co-researchers led a study of the quality and nature of health care with a case study on intergenerational learning about birthing. Post-intervention evaluation: A total of 66 participants were interviewed; the audit of the record was repeated; field notes were kept and observations undertaken in remote sites. Participatory Action Research Study: Baseline data on problems with transfer of information between the regional centre and the remote clinics led to a study by a senior manager and two researchers on improving the system. Costing study: A total of 315 mothers and singleton infants who were clients of the midwifery group practice were compared with 408 mothers with singleton pregnancies from the baseline study post-midwifery group practice intervention. Data on direct costs from the department’s perspective were collected from the first antenatal visit until 6 weeks post-partum, and data on infant costs were collected from birth to 28 days. Benchmarking of neonatal nursery admissions: Records of all 463 neonates born in 2010 and admitted to nursery were benchmarked.	A mixed-methods health services research program of work was designed, using a participatory approach.	The study consisted of two large remote Aboriginal communities in the Top End of Australia and the hospital in the regional centre that provided birth and tertiary care for these communities. The stakeholders included consumers, midwives, doctors, nurses, Aboriginal health workers, managers, policy makers, and support staff. Data were sourced from hospital and health centre records, perinatal data sets and costing data sets, observations of maternal and infant health service delivery and parenting styles, formal and informal interviews with providers and women and focus groups. Studies examined indicator sets that identify best care, the impact of quality of care and remoteness on health outcomes, discrepancies in the birth counts in a range of different data sets and ethnographic studies of ‘out of hospital’ or health-centre birth and parenting. A new model of maternity care was introduced by the health service aiming to improve care following the findings of the research. Some of these improvements introduced during the five-year research program were evaluated.	1 + 1 = A Healthy Start to Life Project. Focus on health services in the year before and the year after birth to promote a healthy start to life. This became the main health-service-led ‘intervention’ of the study.	Overall, sustainable improvements in the maternity services for remote-dwelling Aboriginal women and their infants in the Top End of Australia occurred as a result of the midwifery group practice (MGP) intervention. These included significant improvements in maternal record keeping, antenatal care and screening, smoking cessation advice, a reduction in foetal distress in labour, and a higher proportion of women receiving postnatal contraception advice. Positive experiences of the women and MGP staff were also reported during the first year of the MGP intervention. Continuity of care, provided by appropriately qualified staff as part of the intervention, resulted in improved relationships between the midwives and their clients. The women’s engagement with other health services, facilitated by the midwives, also improved. Additionally, overall costs were reduced as a result of a significant reduction in birthing and neonatal nursery costs as a result of the MGP intervention. However, a review of this intervention conducted in 2012 showed further improvement in clinical care was still needed. Some adverse health conditions appeared to increase, possibly due to improved documentation. Specifically, unacceptable standards of infant care and parental support, no apparent consideration for the fluctuation in numbers and complexity of client cases and adequately trained staff with the required skills for providing care for children in an ‘outpatient’ model of care. Adequate coordination between remote and tertiary services was absent, which is essential to improve quality of care and reduce the risk of poor health outcomes.
Bar-Zeev, S.; Barclay, L.; Farrington, C.; Kildea, S., 2012, [21]. From hospital to home: the quality and safety of a postnatal discharge system used for remote dwelling Aboriginal mothers and infants in the top end of Australia.	A total of 420 women were eligible for the study, sought from 413 medical records at the regional hospital and 400 at the remote health service. A total of 66 semi-structured interviews were conducted with key health and management staff and 30 administrative staff employed in the health centres; 18 staff from the regional hospital maternity, neonatal, and paediatric units; and 12 other staff providing clinical, administrative, or logistical support for remote-dwelling women during pregnancy, around the time of birth, and during the first year of life.	Mixed-methods study, retrospective cohort study, and key informant interviews.	The study aimed to examine the transition of care in the postnatal period from a regional hospital to a remote health service and describe the quality and safety implications for remote-dwelling Aboriginal mothers and their infants.	None introduced.	This study found that there was poor discharge documentation, communication, and co-ordination between the hospital and remote health centre staff. In addition, the lack of clinical governance and a specific position holding responsibility for the postnatal discharge planning process in the hospital system were identified as serious risks to the safety of the mother and infant.
Homer, C.; Foureur, M.; Allende, T.; Pekin, F.; Caplice, S.; Catling-Paull, C., 2012, [22]. ‘It’s more than just having a baby’ women’s experiences of a maternity service for Australian Aboriginal and Torres Strait Islander families.	Clinical outcomes for the 353 women who were booked with the Malabar Community Midwifery Link Service and gave birth in the 2007 and 2008 calendar years were collected prospectively from the database.	Clinical outcomes were collected prospectively and quantitatively analysed. Data from the 353 women who were booked with the Malabar Community Midwifery Link Service were transcribed and analysed qualitatively.	The paper evaluates the Malabar Community Midwifery Link Service from the perspective of Aboriginal women who accessed it.	Malabar Community Midwifery Link Service. The intervention aims to improve maternal and infant health by providing culturally appropriate care. The midwives work closely with the Aboriginal Health Education Officer and in a continuity of care model in which women get to know the midwives during the pregnancy.	Accessing the Malabar Community Midwifery Link Service helped women reduce their smoking during pregnancy. Focus group findings showed that women felt the service provided ease of access, continuity of care, and trust and trusting relationships. A total of 353 women gave birth through accessing the Malabar Community Midwifery Link Service, with forty per cent of babies identified as Aboriginal or Torres Strait Islander. Over ninety per cent of women had their first visit before 20 weeks of pregnancy.
Josif, C.; Kruske, S.; Kildea, S.; Barclay, L., 2017, [23]. The quality of health services provided to remote dwelling Aboriginal infants in the top end of northern Australia following health system changes: a qualitative analysis.	Data were collected from 25 clinicians providing or managing infant health services in the two study sites.	Semi-structured interviews, participant observation, and field notes were analysed thematically.	The study describes infant health service quality following health system changes in the area.	Health system changes. These reforms included implementation of the Healthy Under 5 Kids (HU5K) program and an education package to support staff to deliver this program. Designated Child and Family Health Nurses and Aboriginal Community Worker positions were also established in the two Healthy Start to Life study sites.	A range of factors affecting the quality of care persisted following health system changes in the two study sites. These factors included ineffective service delivery, inadequate staffing, and culturally unsafe practices. The six sub-themes identified in the data, namely, ‘very adhoc’, ‘swallowed by acute’, ‘going under’, ‘a flux’, ‘a huge barrier’, and ‘them and us’, illustrate how these factors continued following health system changes in the two study sites and, when combined, portray a ‘very chaotic system’. Improvements are needed to the quality, cultural responsiveness, and effectiveness of the health services.
McCalman, J.; Searles, A.; Bainbridge, R.; Ham, R.; Mein, J.; Neville, J.; Campbell, S.; Tsey, K., 2015, [11]. Empowering families by engaging and relating Murri way: a grounded theory study of the implementation of the Cape York Baby Basket program.	In-person interviews of 7 women and 3 of their family members who had received Baby Baskets were conducted. The women, aged 21 to 34 years, were either pregnant or recently pregnant and were from six of the eleven indigenous communities in Cape York, Australia. Focus groups were conducted with 18 healthcare workers.	Constructivist-grounded-theory method.	To address the region’s poor maternal and child health, the Baby Basket program was developed by Apunipima Cape York Health Council (ACYHC), a community-controlled Aboriginal health organization located in north Queensland, Australia. The program is an initiative focused on indigenous women who are expecting a baby or have recently given birth.	Apunipima Baby Basket program. Engaging and relating Murri way occurred through four strategies: connecting through practical support, creating a culturally safe practice, becoming informed and informing others, and linking at the clinic.	Overall, the Apunipima Baby Basket program intervention enabled sustainable improvements in the areas of maternal and child health. Engaging and relating Murri way occurred through four strategies: connecting through practical support, creating a culturally safe practice, becoming informed and informing others, and linking at the clinic. These strategies resulted in women and families taking responsibility for health through making healthy choices, becoming empowered health consumers, and advocating for community changes.
Zarnowiecki, D.; Nguyen, H.; Hampton, C.; Boffa, J.; Segal, L., 2018, [24]. The Australian Nurse-Family Partnership Program for aboriginal mothers and babies: Describing client complexity and implications for program delivery.	Australian Nurse-Family Partnership Program data were collected using standardised data forms by the nurses during their antenatal home visits to 276 clients from 2009 to 2015. These data were used to describe client complexity and adversity in relation to demographic and economic characteristics, mental health, and personal safety. Semi-structured interviews with 11 Australian Nurse-Family Partnership Program staff and key stakeholders explored in more depth the nature of client adversity and how this affected program delivery.	Mixed-methods study using Family Partnership Program data and qualitative data collected in semi-structured interviews with Family Partnership Program staff and key stakeholders. Family Partnership Program data were used to describe the characteristics of Family Partnership Program clients.	The Australian Nurse-Family Partnership Program is a home-visiting program for Aboriginal mothers and infants (pregnancy to child’s second birthday) adapted from the United States Nurse Family Partnership program. It aims to improve outcomes for Australian Aboriginal mothers and babies, and disrupt inter-generational cycles of poor health and social and economic disadvantage. The aim of this study was to describe the complexity of Program clients in the Central Australian family partnership program, understand how client complexity affects program delivery, and the implications for desirable program modification.	The Australian Nurse-Family Partnership Program (ANFPP).	Most clients engaged in the Australian Nurse-Family Partnership Program (ANFPP) were described as ‘complicated’, with sixty-six per cent of clients experiencing four or more adversities. These adversities were found challenging for program delivery. For example, housing conditions meant that around half of all ‘home visits’ could not be conducted in the home, being held instead in staff cars or community locations. Extreme poverty, living in insecure housing, and domestic violence (almost one-third of the mothers experiencing more than two episodes of violence in 12 months) affected the delivery of program content and increased the time needed to deliver program content. Additionally, low client literacy meant written handouts were unhelpful for many, requiring the development of pictorial-based program materials. The rates of breastfeeding and child vaccination, which were higher than comparative national data for indigenous women and children in remote areas of Australia, were positive aspects of the ANFPP.

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
