# Peer review of "Models and Interventions to Promote and Support Engagement of First Nations Women with Maternal and Child Health Services: An Integrative Literature Review"

_children, 2022, doi:10.3390/children9050636_

Round 1
Reviewer 1 Report
Thanks for inviting me to review this review article. Generally, this paper has been well written and provided a comprehensive review of the issue regarding the access to maternal and child health services among first nations in Australia. The only thing that I would like to suggest is to add some solid numbers/percentages that showed real discrepancies in the access to maternal and child health services between indigenous and non-indigenous families, if possible. As such, the readers may be in a better standing to understand the scale of this issue, as this journal is aimed at an international community of readers.
Author Response
Reviewer 1:
Thanks for inviting me to review this review article. Generally, this paper has been well written and provided a comprehensive review of the issue regarding the access to maternal and child health services among first nations in Australia. The only thing that I would like to suggest is to add some solid numbers/percentages that showed real discrepancies in the access to maternal and child health services between indigenous and non-indigenous families, if possible. As such, the readers may be in a better standing to understand the scale of this issue, as this journal is aimed at an international community of readers.
Author’s Response:
Thank you for reviewing my article and for your feedback.
I have added this information, which is now lines 63-73. I have made track changes to the original manuscript to reflect this edition. I would be grateful to receive any further feedback if this does not suffice.

Reviewer 2 Report
Thank you for giving me the opportunity to review this interesting work. I have some suggestions to improve the article:
I value positively the barriers that have been identified from the six included studies, which give clues to work on future projects.
In the first paragraph, when they talk about the first years of life, which are the basis for physical, social and emotional life, they would need to add the spiritual dimension (although they name it later on line 194).
It has citation errors. The references are not placed according to the standards of the journal.
Author Response
Reviewer 2:
Thank you for giving me the opportunity to review this interesting work. I have some suggestions to improve the article:
I value positively the barriers that have been identified from the six included studies, which give clues to work on future projects.
In the first paragraph, when they talk about the first years of life, which are the basis for physical, social and emotional life, they would need to add the spiritual dimension (although they name it later on line 194).
It has citation errors. The references are not placed according to the standards of the journal.
Author’s Response:
Thank you for reviewing my article and for your feedback.
I have amended the manuscript to incorporate your suggestions. The word ‘spiritual’ has been included in line 32.
I have edited the manuscript to correct the citation errors, according to the Reference List and Citations Style Guide for MDPI Journals. Apologies for this oversight. I have made track changes to the original manuscript to reflect these editions. I would be grateful to receive any further feedback if this does not suffice.